# SkinDuo^TM^ as a Targeted Probiotic Therapy: Shifts in Skin Microbiota and Clinical Outcomes in Acne Patients

**DOI:** 10.3390/ijms26115000

**Published:** 2025-05-22

**Authors:** Manuele Biazzo, David Pinzauti, Christine Podrini

**Affiliations:** The BioArte Ltd., Malta Life Science Park (LS2.1.10, LS2.1.12-LS2.1.15), Triq San Giljan, SGN 3000 San Gwann, Malta; m.biazzo@thebioarte.com (M.B.); d.pinzauti@thebioarte.com (D.P.)

**Keywords:** acne vulgaris, skin microbiota, *Cutibacterium acnes*, *Staphylococcus epidermidis*, probiotic treatment, *Lactiplantibacillus plantarum*, SkinDuo^TM^

## Abstract

Acne vulgaris is a common dermatological condition strongly associated with disruptions in the skin microbiota, specifically involving key species such as *Cutibacterium acnes* and *Staphylococcus epidermidis*. This study investigates the efficacy of SkinDuo^TM^, a topical probiotic containing *Lactiplantibacillus plantarum*, in modulating the skin microbiota and improving clinical outcomes in patients with acne vulgaris. Over a 4-week to 8-week observational study period, microbial composition and diversity shifts were analyzed using full-length 16S rRNA sequencing. Patient responses were categorized into “good” responders (showing significant clinical improvement) and “no_change” responders (with minimal or no improvement). SkinDuo^TM^ treatment resulted in lower post-treatment *Cutibacterium acnes* abundance in the “good” group compared to the “no_change” group. The “good” group maintained a stable level of alpha diversity following treatment. In contrast, the “no_change” group exhibited a marked reduction in microbial diversity. Beta diversity analysis revealed distinct clustering patterns associated with improved clinical outcomes. These findings suggest that the preservation of microbial richness and evenness may serve as a potential biomarker for positive response to probiotic therapy. This study highlights the potential of SkinDuo^TM^ to restore microbial balance and alleviate acne symptoms, contributing to the growing body of evidence supporting microbiome-based therapeutic strategies in dermatology.

## 1. Introduction

Acne vulgaris is a prevalent chronic inflammatory condition that affects the pilosebaceous units, primarily during adolescence and young adulthood [1]. Traditionally linked to genetic, hormonal, and environmental factors, recent research has highlighted the critical role of the skin microbiota in acne pathogenesis [2,3]. The skin microbiome consists of a dynamic ecosystem of bacteria, fungi, and viruses that contribute to skin homeostasis. Disruptions to this equilibrium, particularly in *Cutibacterium acnes* (*C. acnes*) populations, have been implicated in increased inflammation and acne lesion progression [3].

Emerging evidence underscores the strain-specific behaviour of *C. acnes*, where some strains maintain skin health while others exacerbate inflammation [4,5]. In addition to *C. acnes, Staphylococcus epidermidis* (*S. epidermidis*) has been identified as another key player in skin health. While certain *S. epidermidis* strains contribute to microbial balance, others are associated with inflammation and biofilm formation in acne-prone skin [6]. Recent advancements in microbiome-targeted therapies have introduced probiotics as a promising approach in dermatology. *Lactiplantibacillus plantarum (L.plantarum)*, the active ingredient in SkinDuo^TM^, has demonstrated anti-inflammatory and barrier-enhancing properties in preclinical studies [7]. While in vitro and ex vivo experiments have shown promising results [7], clinical studies exploring its efficacy in modulating skin microbiota and improving acne symptoms remain limited. Unlike oral probiotics, which may exert systemic effects after gastrointestinal transit [8], topical applications directly interact with the skin ecosystem. This allows for localized modulation of microbial communities and barrier functions, offering a targeted approach in dermatological interventions. To date, few clinical studies have evaluated topically applied probiotics [9], particularly those containing viable bacteria designed to be reactivated at the time of use.

This study aimed to evaluate the impact of SkinDuo^TM^ on skin microbiota composition and its therapeutic efficacy in acne patients over a four-week to eight-week observational study. This is among the first observational study to assess a viable, lyophilized probiotic applied topically and reactivated upon contact with water [7], allowing live bacteria to interact directly with the skin microbiome in vivo. Using full-length 16S rRNA sequencing, we characterized shifts in microbial diversity and composition, correlating these changes with clinical outcomes. By providing insights into the interplay between microbial balance and acne pathology, this study highlights the potential of microbiome-based therapies in dermatological care [10].

In this study, we employed full-length 16S rRNA sequencing using Oxford Nanopore Technologies (ONT) Oxford, UK, to achieve species-level taxonomic resolution exceeding 99% accuracy [11]. This surpasses the limitations of traditional short-read methods like Illumina sequencing, which typically provide 70–75% species-level accuracy [11,12]. By using full-length sequencing, we were able to detect subtle microbial shifts critical to understanding the complex interactions within the skin microbiota, particularly in the context of acne vulgaris. To further enhance the accuracy and depth of analysis, data were processed through a comprehensive bioinformatics pipeline, which included rigorous quality control steps to remove low-quality and chimeric reads. High-quality reads were mapped against the Genome Taxonomy Database (GTDB), ensuring precise taxonomic classification [13]. This approach allowed for real-time sequencing and the generation of high-resolution microbial profiles, bridging molecular biology with clinical dermatology and offering a level of insight previously unattainable in dermatological research.

These innovations not only improve the reliability of microbial community characterization but also position this study as a significant contribution to acne research by introducing a clinically tested, reactivatable topical probiotic formulation. By leveraging cutting-edge sequencing technologies and robust data analysis, this work advances our understanding of the microbiome’s role in dermatological conditions and lays the groundwork for microbiome-based therapeutic strategies.

## 2. Results

### 2.1. Patients Recruited and Sequencing Depth

A total of 70 participants were initially enrolled, of whom 34 provided paired pre- and post-treatment samples, resulting in a final cohort for analysis. Across all samples, a total of 2,125,944 sequencing reads were generated. The number of reads per sample ranged from a minimum of 30 to a maximum of 117,662, with an average of 31,731 reads per sample. Rarefaction curves determined that a minimum sequencing depth of 3000 reads per sample was necessary for comprehensive taxonomic profiling. Consequently, eight samples that failed to meet this threshold were excluded from further analysis. This rigorous filtering ensured robust downstream analyses.

Analysis of the achieved sequencing depth (Appendix A) revealed that samples achieving adequate read counts were distributed similarly between the “good” responders and “no_changes” responders as evaluated by certified dermatologists (see Section 2.2), minimizing potential biases introduced by uneven sequencing depth. These measures provided a reliable and robust dataset for microbiota analysis, even though only seven participants provided paired samples pre- and post-treatment for full microbiota analysis.

### 2.2. Clinical Outcomes

Clinical assessments revealed a measurable improvement in acne symptoms among patients treated with SkinDuo^TM^. Participants were categorized into two groups based on clinical outcomes: “good” responders, who showed marked reductions in acne severity, and “no_changes” responders, who exhibited minimal to no clinical improvement. Dermatological evaluations, conducted by certified dermatologists at Cosmetic Derma Medicine, Athens, Greece, utilized standardized visual assessment criteria—including counts of closed and open comedones, papules, and pustules, and evaluations of erythema and sebum production [14]. Improvements in the “good” group were evidenced by reductions in inflammatory lesions and improved skin texture and clarity. In contrast, the “no_changes” group presented persistent acne signs with limited symptom relief across the same clinical markers. These clinical trends were strongly corroborated by corresponding microbiome shifts.

Table 1 presents a detailed summary of clinical evaluations performed before and after the four-to eight-week SkinDuo^TM^ treatments. The outcomes support the therapeutic potential of SkinDuo^TM^ in improving skin health, with associated microbial changes that mirror the degree of clinical response (Figure 1 and Table 1). For each patient, total lesion count was calculated at baseline and after treatment by summing the number of closed comedones, open comedones, papules, and pustules. This composite score was used as a quantitative indicator of acne severity to evaluate treatment response (Appendix A).

### 2.3. Microbial Community Analysis

The study cohort included 34 patients, with 20 in the “good” category and 14 in the “no_changes” category. A total of 17 samples were collected before treatment (“A” samples) and 17 were collected post-treatment (“B” samples). After pre-processing analysis, a total of 6 samples were filtered out (03a, 09b, 21a, 50b, and 66b), as not enough sequencing data were collected (<2000 mapped reads). The resulting phyloseq object included 312 taxa (species-level) and 29 samples organized as follows: 15 samples before treatment “A”, 8 “B_good” (post-treatment, improved skin-derma conditions), and 6 “B_nochange” (post-treatment, no changes detected). To investigate alpha diversity, post-treatment samples were stratified based on clinical outcomes into two subgroups: B_good (patients showing clinical improvement) and B_nochange (patients showing minimal or no improvement). This stratification revealed a clearer pattern in microbial diversity (Figure 2). Alpha diversity, measured by three indices—Observed Species, the Shannon Index, and the Simpson Index—was significantly lower in the B_nochange group compared to both Group A (pre-treatment) and Group B_good (post-treatment with improvement) (*p* < 0.05). These differences were especially pronounced in the Shannon and Simpson indices, which account for both richness and evenness. In contrast, no significant differences in alpha diversity were observed between Group A and B_good, suggesting that microbial richness and community evenness were maintained in patients who responded positively to SkinDuo^TM^ treatment (Figure 2, middle and right panels).

The decreased diversity observed in the B_nochange group may point to a destabilized or dysbiotic microbial environment that lacks the resilience necessary to support skin health and respond to treatment. This lower diversity could reduce the ecological buffering capacity of the skin microbiome, making it less adaptable and more prone to imbalance. On the other hand, the preserved microbial diversity in the B_good group suggests a potentially more robust and functionally stable microbiota that can interact beneficially with host skin processes. These findings underscore the possibility that alpha diversity itself may be a biomarker of therapeutic response, highlighting the need to consider individual microbial profiles in future personalized skin microbiome therapies. In summary, the analysis included 34 patients, subdivided into 20 responders (B_good) and 14 non-responders (B_nochange), with 29 samples retained after quality filtering. The B_good group maintained a stable level of alpha diversity following treatment, whereas the B_nochange group exhibited a marked reduction. These findings suggest that the preservation of microbial richness and evenness may serve as a potential biomarker for clinical response to probiotic therapy.

### 2.4. Longitudinal Changes by Clinical Response

To further examine changes in skin microbiota in response to SkinDuo^TM^ treatment, we analyzed alpha diversity stratified not only by timepoint (pre- vs. post-treatment) but also by clinical outcome (good vs. no change). Figure 3 presents alpha diversity indices (Observed Species, Shannon, and Simpson) across paired samples before and after treatment in patients with good clinical outcomes versus those without improvement.

In patients who experienced improvements (B_good), alpha diversity indices remained stable or showed mild decreases after treatment, with no statistically significant differences in Observed Species (Wilcoxon *p* = 0.17), the Shannon Index (*p* = 0.066), or the Simpson Index (*p* = 0.05) in post-treatment samples. This indicates that microbial richness and evenness were largely preserved in responders, supporting the notion that SkinDuo^TM^ helped maintain a balanced skin microbiome conducive to clinical benefit.

Conversely, in the no-change group (B_nochange), significant reductions in alpha diversity were observed post-treatment. Compared to their baseline (A), the B_nochange group exhibited decreased Observed Species (*p* = 0.028), Shannon Diversity (*p* = 0.029), and Simpson Diversity (*p* = 0.043). These findings suggest that in non-responders, SkinDuo^TM^ application may have contributed to a decline in microbial diversity, potentially reflecting dysbiosis or a loss of beneficial microbial members.

Altogether, these results reinforce a key distinction between responders and non-responders to SkinDuo^TM^ treatment: while successful outcomes were associated with microbial stability, clinical non-improvement was paralleled by a reduction in skin microbial diversity.

### 2.5. Microbial Composition Shifts and Key Taxa Indicators

To further characterize the relationship between microbial community composition and clinical response, a Principal Coordinate Analysis (PCoA) was performed using Bray–Curtis dissimilarity, with species-level taxa projected as vectors. As shown in Figure 4, the first coordinate (PCoA1) explained 30.82% of the total variance, while PCoA2 and PCoA3 accounted for 15.59% and 10.5%, respectively. Samples from the B_good group (blue circles) clustered more tightly within the 95% confidence ellipse, suggesting greater microbial convergence following successful SkinDuo^TM^ treatment. In contrast, samples from the B_nochange group (blue squares) were more dispersed, indicating higher compositional heterogeneity (Figure 4). Permanova analysis (adonis2, vegan package) yielded statistically significant differences (*p* = 0.012).

Taxonomic projections revealed that *Cutibacterium acnes* and *Streptococcus salivarius* were oriented toward the B_good cluster, suggesting a potential association with favorable outcomes. Conversely, taxa such as *Staphylococcus epidermidis, Klebsiella michiganensis*, and *Pseudomonas nitroreducens* appeared more associated with B_nochange samples, potentially reflecting microbial instability or pathogenic overgrowth. While these vectors should be interpreted cautiously and do not establish causality, they offer insight into taxa that may contribute to or reflect divergent microbiota states between responders and non-responders.

To further explore compositional changes associated with clinical response, differential abundance analysis (DAA) was conducted using the ANCOM-BC method (Figure 5). This analysis evaluated log-fold changes (LFCs) in species abundance for both the B_good and B_nochange groups, each compared to their own pre-treatment baseline (Group A). Although the ANCOM-BC tool did not detect any taxa as significantly differentially abundant in a direct comparison between B_good and B_nochange, several species exhibited statistically significant changes when each post-treatment group was compared individually to the baseline. For example, in the B_good group, notable decreases were observed in *Streptococcus mitis* (q < 0.001), QFNR01 sp016864895 (q < 0.05), and *Staphylococcus debuckii* (q < 0.001), while *Rothia mucilaginosa* and *Staphylococcus capitis* increased in abundance. These shifts may indicate beneficial remodeling of skin microbiota in treatment responders. In contrast, the B_nochange group showed significant increases in species such as *Roseomonas gilardii* (q < 0.001), *Ralstonia sp001078575* (q < 0.001), and *Staphylococcus capitis*, coupled with decreases in *Staphylococcus debuckii* (q < 0.01) and *Streptococcus intermedius* (q < 0.001). This pattern may suggest a destabilized or dysbiotic microbiota trajectory among non-responders.

### 2.6. Taxonomic Shifts Based on Core Microbiota

Taxonomic profiling at both the phylum and species levels was conducted after applying a 30% prevalence threshold to identify core microbiota. As illustrated in Figure 6, *Actinobacteriota* remained the dominant phylum across all groups, with its relative abundance increasing progressively from Group A to B_good and peaking in B_nochange. In contrast, members of the *Firmicutes* and *Proteobacteria* phyla were more variable and showed notable reduction in B_good and B_nochange compared to pre-treatment samples.

At the species level (Figure 7), *Cutibacterium acnes* was the most abundant across all groups, with the highest average relative abundance observed in the B_nochange group (96.0%) and slightly lower levels in the B_good (89.4%) and A (86.8%) groups. Interestingly, *Staphylococcus epidermidis*, a commonly associated skin commensal, was markedly reduced in B_nochange (0.8%) compared to B_good (7.4%) and A (6.5%). The taxon QFNR01 sp016864895, potentially linked to non-skin microbial origins [15], showed variable presence—higher in A and B_nochange (~2.9–3.5%) and nearly absent in B_good (0.5%). Notably, *Lactiplantibacillus plantarum*, the probiotic species applied in SkinDuo^TM^, was detected at higher levels in B_good (1.5%) compared to A (0.16%) and was absent in B_nochange, suggesting a potential engraftment or transient colonization in responders.

Other species, such as *Lawsonella clevelandensis* and *Streptococcus oralis*, also showed decreased abundance in B_nochange, aligning with the pattern of reduced microbial diversity and complexity in non-responders. These patterns support the hypothesis that SkinDuo^TM^ treatment promotes stabilization of a beneficial skin microbiota profile, with reductions in opportunistic taxa and increased colonization by targeted probiotic strains in individuals exhibiting clinical improvement.

To complement the taxonomic composition analysis, we calculated the average relative abundance of key species across treatment groups. Table 2 summarizes the relative abundances observed in the pre-treatment (A), post-treatment with clinical improvement (B_good), and post-treatment with no improvement (B_nochange) groups:

These values indicate that *Cutibacterium acnes* remained the most abundant species across all groups, with a noticeable increase in the B_nochange group. Interestingly, *Lactiplantibacillus plantarum* was significantly enriched in the B_good group, aligning with the probiotic nature of SkinDuo^TM^. In contrast, potential indicators of dysbiosis, such as QFNR01 sp016864895 and *Streptococcus oralis*, showed reduced abundance in B_good compared to A and B_nochange.

These quantitative shifts in microbial abundance provide additional context to the alpha and beta diversity results, suggesting that clinical improvement may be linked not only to overall diversity but also to specific taxonomic compositions.

## 3. Discussion

This study evaluated the impact of the topical probiotic formulation SkinDuo^TM^ on the skin microbiota and clinical outcomes in patients with acne vulgaris. Through a multidimensional approach that included alpha and beta diversity analysis, taxonomic profiling, and stratification by clinical response, our results demonstrate a strong association between microbial community shifts and dermatological improvement.

The use of probiotics in dermatology, particularly for conditions like acne [16], has gained increasing attention. SkinDuo^TM^ contains *Lactiplantibacillus plantarum*, a probiotic species with demonstrated anti-inflammatory and antimicrobial effects. Our findings align with the results of Podrini et al. [7], who showed that *L. plantarum* maintained viability on human skin and significantly reduced lipid production and inflammatory markers in sebocyte cultures infected with virulent acne-associated bacteria. In our clinical cohort, *L. plantarum* was found in higher relative abundance in the B_good group, suggesting transient colonization or probiotic-induced modulation in responders.

A central finding was the divergence in alpha diversity patterns between responders and non-responders. The B_good group maintained microbial richness and evenness post-treatment, indicating a resilient microbial community. In contrast, the B_nochange group experienced a statistically significant decline across the Shannon and Simpson Diversity indices, suggesting microbial instability or collapse. These findings reinforce the hypothesis that diversity and ecological balance are essential to treatment response—echoing insights from recent reviews by Niedźwiedzka et al. [2], who highlighted microbial diversity as a therapeutic target in acne management.

Beta diversity analysis further confirmed the stratified effect of treatment. Post-treatment samples from the B_good group formed a more cohesive cluster, implying microbial convergence and stabilization. In contrast, B_nochange samples were more heterogeneous, suggesting an unstable or variable community composition. Taxonomic vectors in PCoA plots pointed to an increased abundance of beneficial taxa like *Cutibacterium acnes* and *Lactiplantibacillus plantarum* in responders, and an elevated presence of potentially less favorable taxa like *Pseudomonas nitroreducens* and *Klebsiella michiganensis* in non-responders.

The taxonomic composition at the species level revealed specific shifts. While *C. acnes* remained dominant across all groups, its abundance was highest in the B_nochange group. However, this does not necessarily denote a pathogenic role, as *C. acnes* comprises multiple strains with varying effects. In contrast, *Staphylococcus epidermidis*, often considered a protective commensal, was reduced in B_nochange, potentially implicating a disrupted balance. The sharp reduction in *L. plantarum* in the B_nochange group aligns with the hypothesis that engraftment or activity of the probiotic is necessary for clinical benefit. This observation raises the possibility that non-responders may harbor intrinsic barriers to probiotic engraftment. Factors such as compromised skin barrier function, heightened local immune activity, or a dysbiotic baseline microbiota less receptive to colonization could play a role. Alternatively, the detectable signal in responders may reflect a transient colonization or functional modulation by the probiotic strain, rather than permanent engraftment. These hypotheses warrant further investigation to better understand the prerequisites for effective probiotic action on the skin.

Differential abundance analysis using ANCOM-BC did not yield statistically significant differentially abundant taxa when comparing B_good to B_nochange directly. However, when both groups were analyzed relative to the pre-treatment baseline (Group A), log2 fold-change plots revealed distinctive trends. Beneficial taxa like *Rothia mucilaginosa* [17] and *L. plantarum* [18] were enriched in B_good, whereas opportunistic or inflammation-associated species such as *Roseomonas gilardii* [19] and *Ralstonia* [20] were enriched in B_nochange. These observations underscore the subtle but biologically relevant shifts in microbial profiles associated with clinical response.

Moreover, this study benefitted from high-resolution, full-length 16S rRNA sequencing using Oxford Nanopore technology, allowing accurate species-level identification and capturing taxa often missed by short-read methods [11]. This advanced approach enhanced our ability to detect meaningful shifts in the skin microbiome, positioning this methodology as a valuable tool in dermatological microbiome research.

Despite these insights, the study has limitations. The small cohort size reduces statistical power and generalizability. Additionally, while relative abundance and diversity metrics offer valuable perspectives, functional profiling of microbial activity would provide more mechanistic insights. Future studies incorporating metagenomic or metabolomic analyses could validate and expand on these findings.

In conclusion, this study supports the use of targeted probiotics like SkinDuo^TM^ as adjunctive treatments for acne. The results highlight the role of microbiota modulation in therapeutic response and suggest that microbial diversity and specific taxa shifts could serve as biomarkers of treatment efficacy. However, the variability in patient response underscores the need for personalized microbiome-based approaches in dermatological care. Overall, these findings position SkinDuo^TM^ as a promising adjunct in personalized dermatological protocols targeting acne. The consistent shifts in microbial composition among responders support the use of microbiota-based strategies as a complement to conventional treatments. This highlights the relevance of microbiome-informed dermatology, in which microbial baseline profiling could guide individualized therapeutic choices.

### Limitations and Future Directions

While this study provides valuable insights into the therapeutic potential of the SkinDuo^TM^ probiotic, several limitations should be noted. First, the sample size was relatively small (n = 29 after filtering), which may limit the statistical power and generalizability of the findings. Similar limitations have been noted in prior microbiome-focused dermatological studies, such as Thiruppathy et al. [21], where variability in host responses and small cohorts impeded robust conclusions about treatment efficacy across broader populations. While short-term studies have explored the impact of probiotics on skin microbiomes in acne patients, there is a paucity of longitudinal research assessing sustained or delayed microbiome shifts. This underscores the need for extended monitoring to fully understand the long-term effects of probiotic interventions.

No placebo or vehicle-only control group was included in this study. This omission limits our ability to distinguish the specific effects of the SkinDuo^TM^ formulation from general skincare benefits or placebo responses. Future trials should incorporate appropriate control arms to isolate probiotic-driven outcomes and strengthen causal inferences.

Another key limitation is the reliance on 16S rRNA amplicon sequencing, which, while informative for community structure, lacks functional resolution and may overlook strain-level differences that are often critical in probiotic research [22]. Future studies employing shotgun metagenomics or transcriptomics could provide a more comprehensive view of microbial function and metabolic contributions to skin health. Lastly, while *Lactiplantibacillus plantarum* appeared to increase in abundance in responders, causal mechanisms of action remain speculative without experimental validation or host–microbe interaction studies.

Despite these limitations, this pilot investigation supports the feasibility of using topical probiotics to modulate the skin microbiome and improve clinical outcomes in acne. The study demonstrates the practical integration of full-length 16S rRNA sequencing and targeted probiotic application into dermatological research, establishing a solid methodological foundation. Larger, multi-center studies with diverse populations and longer intervention windows are necessary to fully validate and expand upon these findings.

## 4. Materials and Methods

### 4.1. Study Design and Participants

This longitudinal, in vivo study was designed to evaluate the effects of SkinDuo^TM^, a topical probiotic containing *Lactiplantibacillus plantarum*, on the skin microbiota of patients diagnosed with acne vulgaris. Conducted in adherence to the Declaration of Helsinki, the study was approved by the IMAS Institute of Microbiome and Applied Science at Malta Life Science Park (LS2.1.12-LS2.1.15), Triq San Giljan, San Gwann, SGN 3000 Malta under approval code RS0001. The recruitment of participants occurred through dermatology clinics at Cosmetic Derma Medicine, LEOF Kifissias 252, Athens, Greece, with patients required to meet specific inclusion and exclusion criteria.

Participants were eligible for inclusion if they had a clinical diagnosis of moderate-to-severe acne and had not used systemic antibiotics or probiotics within six months prior to enrollment. Exclusion criteria included pregnancy, known allergies to probiotics, or concurrent dermatological treatments. Prior to participation, all patients provided written informed consent, which covered the collection of skin swabs, as well as the capture of standardized photographs documenting their skin condition.

To control for potential confounding factors affecting the skin microbiome, participants were instructed to maintain consistent dietary habits and avoid introducing new oral or topical treatments during the study. Use of cosmetic products was standardized across all subjects to a provided non-comedogenic cleanser. Lifestyle factors were also collected via questionnaire (Appendix A).

Baseline demographic data, including age, gender (Appendix A), and acne severity scores based on the Global Acne Grading System (GAGS), were recorded to ensure balanced representation across the study population. Follow-up data were collected 4 and 8 weeks after treatment to assess changes in skin condition and microbiota composition (Table 1).

### 4.2. SkinDuo^TM^ Treatment Protocol

Participants applied SkinDuo^TM^, a topical probiotic serum containing *Lactiplantibacillus plantarum* (2 × 10^8^ CFU/mL), once daily at bedtime. The treatment was self-administered following an initial demonstration by a dermatologist to ensure proper application technique. Participants were instructed to avoid applying other new topical or oral treatments during the study period to isolate the effects of SkinDuo^TM^. A standardized skincare regimen was maintained across all participants, including gentle cleansing with a non-comedogenic cleanser. Compliance was monitored through bi-weekly check-ins, and participants maintained logs to record application adherence and any adverse events.

### 4.3. Sample Collection and Processing

Skin swabs, CuDerm (D100—D-Squame Standard Sampling Discs Clinical and Derm Dallas, TX, USA), were collected from the primary areas affected by acne. Samples were collected under standardized conditions by the dermatologists to minimize variability.

At each sampling session (baseline and post-treatment), the dermatologist assessed skin condition, noting clinical parameters such as comedone count, pustule presence, and erythema. CuDerms were transferred immediately into sterile tubes and stored at −20 °C until DNA extraction. DNA was extracted from skin swab samples with the MagMax Microbiome Ultra Nucleic Acid Isolation Kit (Applied Biosystems, Carlsbad, CA, USA), in combination with a KingFisher Flex Purification System (Thermo Fisher Scientific, Waltham, MA, USA). The samples were lysed using the homogenizer MP FastPre-24 5G (MP Biomedical, Irvine, CA, USA), relying on a bead-beating approach (mechanical lysis). DNA extraction was performed according to the manufacturer’s instruction. Isolated DNA was quantified with the Qubit 4 Fluorometer (Thermo Fisher Scientific) using the dsDNA high sensitivity (HS) kit (Thermo Fisher Scientific, Waltham, MA, USA).

### 4.4. Microbiota Analysis and Sequencing

To characterize the skin microbiota, we employed full-length 16S rRNA gene sequencing using Oxford Nanopore Technologies (ONT), which enables high-resolution taxonomic classification.

The full-length 16S rRNA gene (~1500 bp in length) was amplified with the primers 27f (5′-TTTCTGTTGGTGCTGATATTGC-AGRGTTYGATYMTGGCTCAG-3′) and 1492r (5′-ACTTGCCTGTCGCTCTATCTTC-CGGTTACCTTGTTACGACTT-3′) as previously described [12,13], introducing a few modifications. The PCR reaction was carried out in a 25 µL total volume containing 12.5 µL Q5 Hot-Start High Fidelity 2× Master Mix (New England Biolabs) (New England Biolabs, Ipswich, MA, USA), the primers (400 nM), and 8.5 µL of template DNA. The reaction was run in a T100 thermal cycler (BioRad, Hercules, CA, USA) using the following program: initial denaturation at 95 °C for 3 min; followed by 32 cycles at 95 °C for 20 s, 55 °C for 30 s, and 72 °C for 2 min; and a final extension at 72 °C (5 min). The PCR products were checked on a 2% agarose gel and cleaned with Agencourt AMPure XP beads (Beckman Coulter, Indianapolis, IN, USA). Briefly, 0.6× AMPure beads were added to the reaction, incubating for 5 min at room temperature. Beads were pelleted using a magnetic rack (NimaGen, Nijmegen, The Netherlands), washed twice with freshly prepared 70% ethanol, and resuspended in 15 µL of nuclease-free water. After 10 min of incubation at room temperature, beads were pelleted again in the magnetic rack and the eluate was collected. Yields and purity values were measured using a NanoDrop 8000 spectrophotometer (Thermo Fisher Scientific).

Each sample was associated with a unique molecular barcode sequence, enabling sample multiplexing. Molecular barcodes were added in a second PCR reaction, using a modified version of the “PCR Barcoding Expansion 1–96 kit” (Oxford Nanopore Technologies (ONT), Oxford, UK). A customized panel of 5′-phosphorylated primers was used to skip the End-Prep and dA tailing reactions (ONT), as previously described [12,13].

The barcoded amplicons were pooled together in a 30 µL volume containing 500 ng DNA. Sequencing adapters were ligated using the ligation sequencing kit SQK-LSK114 (ONT, Oxford, UK). In short, 12.5 µL of Ligation Buffer (LNB), 5 µL of Quick T4 Ligase, and 2.5 µL of Ligation Adapter (LA) were added to the 30 µL pool. The reaction was incubated at room temperature for 10 min. AMPure XP beads (0.4×) were added to the reaction and incubated for 5 min. The beads were pelleted on a magnetic rack and washed twice with 250 µL of Short Fragment Buffer (SFB). The bead pellet was then resuspended in 15 µL of elution buffer, incubated for 10 min at room temperature, and pelleted again in the magnetic rack. After pelleting, the eluate was carefully transferred into a clean Eppendorf DNA LoBind tube. Flow cell priming and library loading were performed according to the manufacturer’s instructions: 10 fmol of DNA library was gently loaded in a dropwise manner onto a R10.4.1 flow cell (ONT). The sequencing run was performed using a GridION Mk1b device (ONT), enabling real-time super-accurate basecalling and live demultiplexing (using default parameters).

### 4.5. Bioinformatics and Data Processing

Raw sequencing reads were processed using a custom bioinformatics pipeline designed for high-accuracy microbial community analysis. The pipeline consisted of the following steps. An initial pre-processing step was performed, removing low-quality and chimeric reads and retaining reads with a length between 1800 and 1200 nucleotides in length. Next, emu tool v.3.4.4 [23] was used to infer microbial taxonomy, mapping reads against a manually curated reference database, the Genome Taxonomy Database (GTDB, Release V.207), which was downloaded from the official repository (https://data.gtdb.ecogenomic.org/releases/, accessed on 1 March 2023 as described previously [13]. Taxonomy tables were finally imported into RStudio (build v. 2023.12.1, R v.4.3.2) from downstream community analysis. Samples with less than 2000 mapped reads were removed from the dataset. The presence of contaminants was assessed by looking at negative samples. Since we hypothesized a putative cross-contamination between skin and gut samples, the following microbial species were filtered out: *R. bromii*, *S. enterica*, *B. pseudo-catenulatum*, *B. wexlerae*, *E. coli*, *F. prausnitzii*, *A. rectalis*, *P. vulgatus*, and *P. copri*. Alpha diversity analysis was performed using the vegan v 2.6.4 package [Oksanen, J.; Simpson, G.; Blanchet, F.; Kindt, R.; Legendre, P.; Minchin, P.; O’Hara, R.; Solymos, P.; Stevens, M.; Szoecs, E.; et al. Vegan: Community Ecology Package. 2022. Available online: https://CRAN.R-project.org/package=vegan (accessed on 1 March 2023) [24]. Statistical comparisons of alpha diversity between pre- and post-treatment samples, as well as between the “good” and “no_changes” response groups, were performed using the Wilcoxon rank-sum test to identify significant differences in microbial richness and evenness. Beta diversity, or between-sample diversity, was performed using Non-Metric Multidimensional Scaling (NMDS) and Principal Coordinate Analysis (PCoA, using the Bray–Curtis dissimilarity index. The significance of clustering patterns was tested with Permutational Multivariate Analysis of Variance (PERMANOVA) using the adonis2 function from vegan v 2.6.4. Relative abundances of key microbial taxa, including *Cutibacterium acnes* and *Staphylococcus epidermidis*, were compared across cohorts using differential abundance analysis with the ANCOMBC v 2.4.0 package (p_adj_method = “fdr”) [25]. To visualize shared and unique microbial species between the “good” and “no_changes” response groups, an UpSet plot was created using UpSetR (v.1.4.0).

## 5. Patents

MB has a patent submitted in relation to this study: PCT/EP2023/078874.

## Figures and Tables

**Figure 1 ijms-26-05000-f001:**
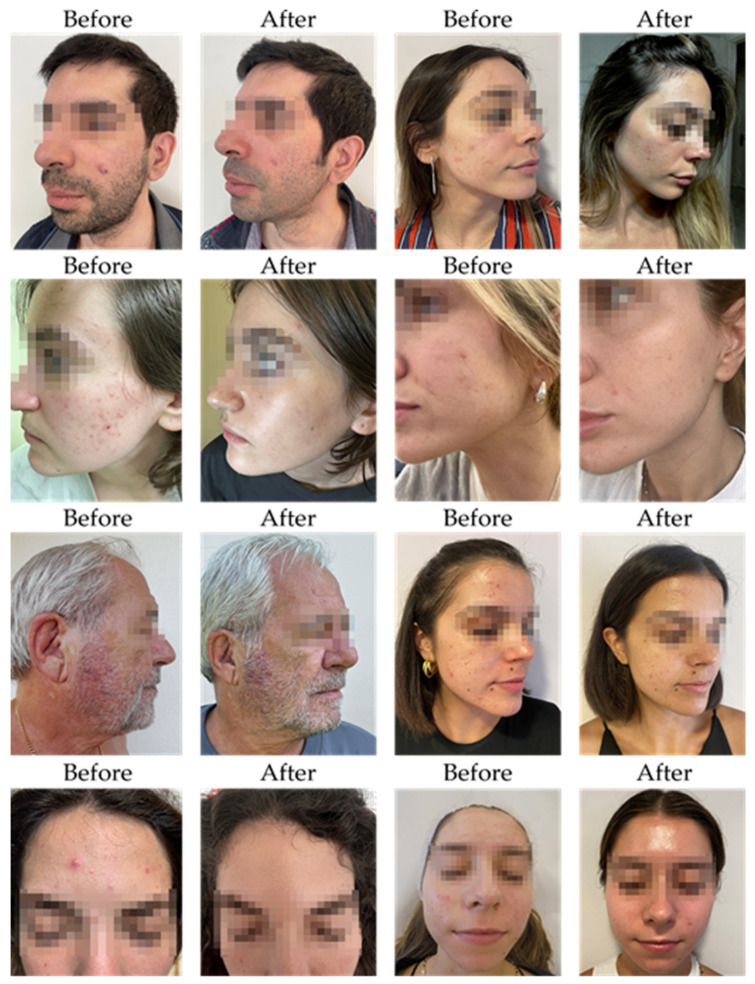
Clinical outcomes before and after SkinDuo^TM^ treatment. Standardized photographic documentation of patients diagnosed with acne vulgaris was captured before (**left**) and after (**right**) 4, 5, or 8 weeks of daily application of SkinDuo^TM^, as per Table 1. Improvements in skin texture, a reduction in inflammation, and overall enhancement of skin health were observed in multiple cases, as assessed by dermatologists. The treatment led to visible reductions in acne lesions, erythema, and overall skin irregularities, highlighting the potential of SkinDuo^TM^ in promoting microbiome balance and dermatological health.

**Figure 2 ijms-26-05000-f002:**
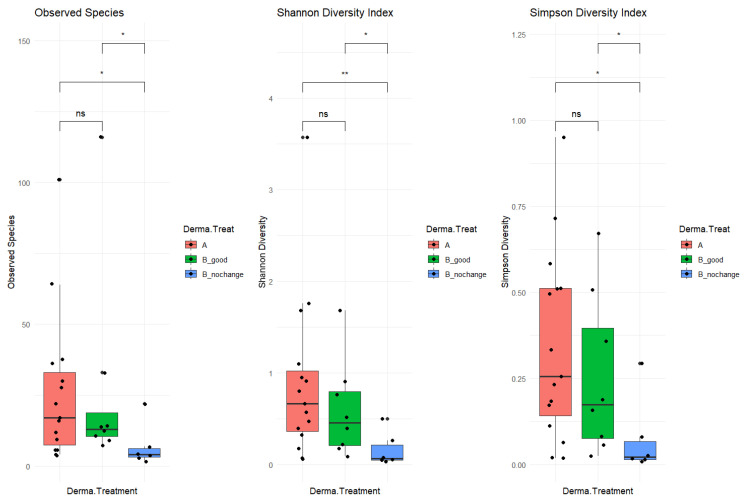
Alpha diversity comparison before and after SkinDuoTM^TM^ treatment with dermatological outcome. Boxplots display the alpha diversity of skin microbiota across three cohorts: Group A (pre-treatment, red), Group B_good (post-treatment, patients with dermatological improvement, green), and Group B_nochange (post-treatment, patients with no improvement, blue). Diversity was assessed using (**left**) Observed Species, (**center**) the Shannon Diversity Index, and (**right**) the Simpson Diversity Index. Statistical comparisons were performed using the Wilcoxon rank-sum test. Significant reductions in diversity were observed in the B_nochange group compared to both Group A and B_good, as indicated by *p*-values < 0.05 (* *p* < 0.05; ** *p* < 0.01, ns means not significant). Specifically, both the Shannon and Simpson indices showed significantly lower diversity in the B_nochange group compared to B_good, highlighting potential microbial instability or dysbiosis in patients who did not respond to SkinDuo^TM^ treatment.

**Figure 3 ijms-26-05000-f003:**
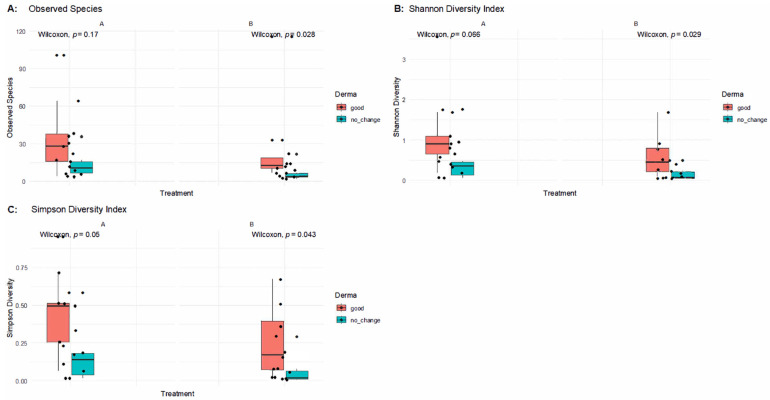
Alpha diversity comparison after SkinDuo^TM^ treatment with dermatological outcome. Stratified analysis of skin alpha diversity by treatment and clinical response. (**A**) Observed Species; (**B**) Shannon Diversity Index; (**C**) Simpson Diversity Index. Red = good clinical response; blue = no_change. Wilcoxon rank-sum test *p*-values shown above each comparison: Observed Species, A vs. B_good (*p* = 0.17), B_good vs. B_nochange (*p* = 0.028); Shannon Index, A vs. B_nochange (*p* = 0.066), B_good vs. B_nochange (*p* = 0.029); Simpson Index, A vs. B_nochange (*p* = 0.05), B_good vs. B_nochange (*p* = 0.043).

**Figure 4 ijms-26-05000-f004:**
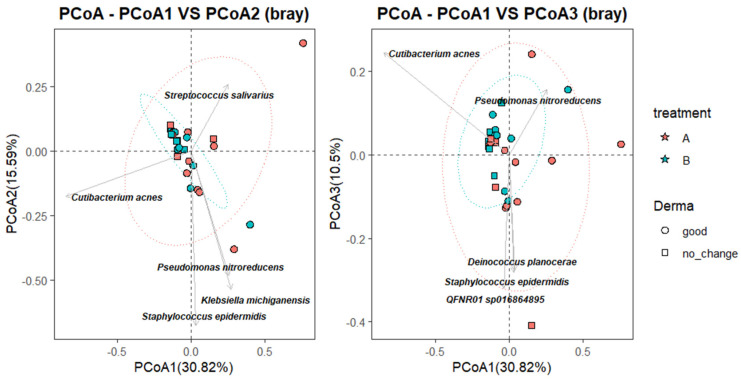
Bray–Curtis Principal Coordinate Analysis (PCoA) of skin microbiota. (**Left**) PCoA1 vs. PCoA2 and (**Right**) PCoA1 vs. PCoA3. Samples are colored by treatment group (A = pre-treatment, circles; B = post-treatment, stars) and shaped by dermatological outcome (good = circle; no_change = square). Ellipses represent 95% confidence intervals for each subgroup. Labeled microbial taxa represents those contributing most strongly to ordination variation, as inferred from vector projections.

**Figure 5 ijms-26-05000-f005:**
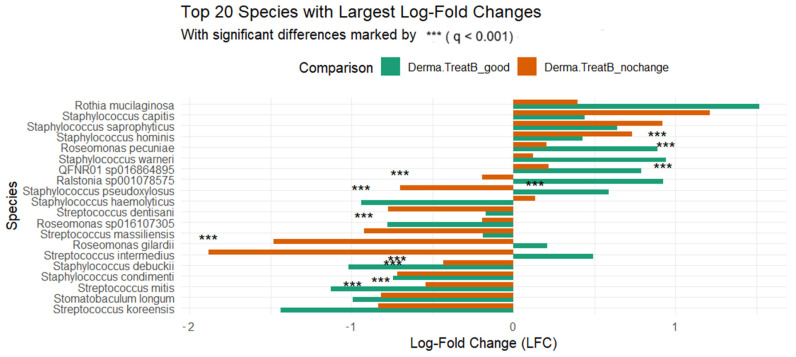
The top 20 taxa with the most pronounced shifts, with statistical significance denoted by adjusted *p*-values (q-values): *** (q < 0.001).

**Figure 6 ijms-26-05000-f006:**
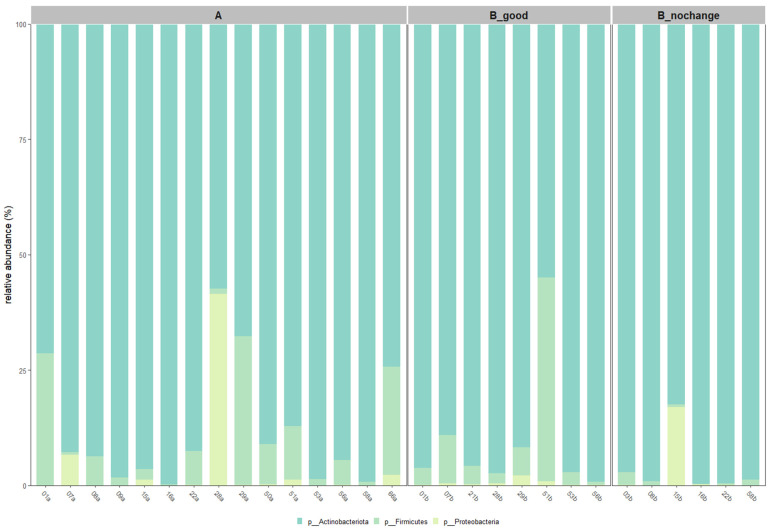
Taxonomic composition of skin microbiota filtered at 30% prevalence threshold. Top: relative abundances of dominant phyla (*Actinobacteriota, Firmicutes, Proteobacteria*) in each individual, grouped by treatment response (A = pre-treatment; B_good = post-treatment with improvement; B_nochange = post-treatment with no improvement).

**Figure 7 ijms-26-05000-f007:**
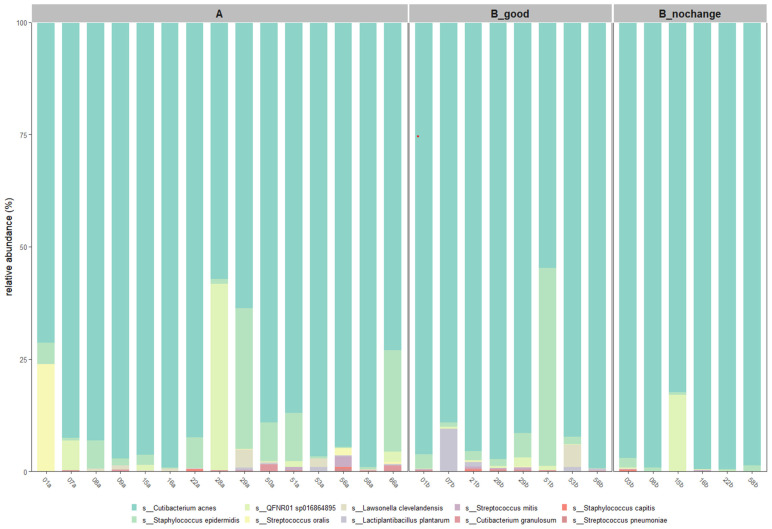
Species-level composition of skin microbiota filtered at 30% prevalence threshold. Species-level composition highlights *Cutibacterium acnes* as the most prevalent taxon across all groups. Increased *Lactiplantibacillus plantarum* levels were observed in B_good, while *Staphylococcus epidermidis* abundance declined in B_nochange. Each bar represents an individual sample, grouped by treatment cohort (A = pre-treatment; B_good = post-treatment with improvement; B_nochange = post-treatment with no improvement).

**Table 1 ijms-26-05000-t001:** Acne grade assessment: before treatment Time 0 and after treatment shown in weeks.

Sample. ID	Treatment	Derma	Derma. Treat	Closed. Comodones	Open. Comodones	Papules	Pustules	Sebum. Production	Erythema	Weeks
01a	A	good	A	5	5	6	2	moderate	mild	0
01b	B	good	B_good	3	3	2	0	mild	absent	4
03a	A	no_change	A	6	7	6	1	mild	mild	0
03b	B	no_change	B_nochange	6	2	6	1	mild	absent	5
07a	A	good	A	5	10	4	0	mild	mild	0
07b	B	good	B_good	0	2	3	0	mild	absent	5
08a	A	no_change	A	5	3	6	0	severe	moderate	0
08b	B	no_change	B_nochange	4	4	4	0	mild	mild	4
09a	A	no_change	A	6	5	10	2	moderate	mild	0
09b	B	no_change	B_nochange	3	3	6	0	mild	absent	4
15a	A	no_change	A	5	2	9	5	moderate	moderate	0
15b	B	no_change	B_nochange	5	3	9	6	moderate	moderate	4
16a	A	no_change	A	0	6	2	0	severe	mild	0
16b	B	no_change	B_nochange	0	4	2	0	mild	mild	4
21a	A	good	A	5	5	4	0	moderate	moderate	0
21b	B	good	B_good	0	0	2	0	absent	mild	4
22a	A	no_change	A	7	8	15	6	moderate	moderate	0
22b	B	no_change	B_nochange	5	4	20	8	moderate	severe	4
28a	A	good	A	30	10	4	0	mild	mild	0
28b	B	good	B_good	10	0	0	0	mild	absent	4
29a	A	good	A	10	6	7	2	mild	mild	0
29b	B	good	B_good	3	5	3	1	mild	mild	4
50a	A	good	A	8	5	4	1	mild	moderate	0
50b	B	good	B_good	0	0	0	0	absent	mild	4
51a	A	good	A	5	4	10	3	severe	moderate	0
51b	B	good	B_good	0	4	6	0	mild	mild	8
53a	A	good	A	7	20	8	3	moderate	mild	0
53b	B	good	B_good	0	10	2	0	mild	absent	4
56a	A	good	A	20	4	6	5	moderate	mild	0
56b	B	good	B_good	5	0	7	1	absent	mild	4
58a	A	no_change	A	6	12	5	7	moderate	moderate	0
58b	B	no_change	B_nochange	10	4	4	5	mild	mild	4
66a	A	good	A	3	5	4	4	mild	mild	0
66b	B	good	B_good	3	0	4	0	mild	absent	4

**Table 2 ijms-26-05000-t002:** Taxonomic table of key species.

Species	A	B_Good	B_Nochange
QFNR01 sp016864895	0.035	0.005	0.029
*Staphylococcus capitis*	0.001	0.001	0.000
*Staphylococcus epidermidis*	0.065	0.074	0.008
*Streptococcus mitis*	0.002	0.002	0.000
*Streptococcus oralis*	0.017	0.000	0.000
*Streptococcus pneumoniae*	0.001	0.001	0.000
*Cutibacterium acnes*	0.868	0.894	0.960
*Cutibacterium granulosum*	0.003	0.001	0.000
*Lactiplantibacillus plantarum*	0.002	0.015	0.000
*Lawsonella clevelandensis*	0.006	0.006	0.001

## Data Availability

The sequencing datasets generated and analyzed in this study have been deposited in the European Nucleotide Archive (ENA) under accession number PRJEB87224. The raw reads related to the SkinDuo^TM^ study are publicly available via this repository.

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
