# Peer review of "SkinDuo^TM^ as a Targeted Probiotic Therapy: Shifts in Skin Microbiota and Clinical Outcomes in Acne Patients"

_ijms, 2025, doi:10.3390/ijms26115000_

Round 1
Reviewer 1 Report
Comments and Suggestions for Authors
General and specific comments
This work deals with the evaluation of a topical probiotic containing Lactiplantibacillus plantarum, in modulating the skin microbiota and improving clinical outcomes in patients with acne vulgaris. The manuscript is relevant for the development of specific cosmetic products aimed at balancing the skin microbiome in acne.
Although the sample size is small, the research is of interest for understanding the use of topical probiotics.
The experimental design is appropriate to test the hypothesis, and the figures, diagrams, and schemes are also suitable. The conclusions are consistent with the evidence and arguments presented; and the references are relevant.
Nevertheless, a minor revision should be made in order to be approved.
The composition of the topical probiotic serum containing Lactiplantibacillus plantarum needs to be better explained. Are the Lactobacillus alive? Or have they been subjected to some process of tyndallization or similar? Since live bacteria cannot be found in cosmetics, further explanation is needed to understand this issue.
Author Response
We thank the reviewer for their thoughtful and constructive comments. We appreciate the recognition of the study’s relevance and scientific merit.
Regarding the specific question about the composition of the topical probiotic serum:
The product contains Lactiplantibacillus plantarum (SkinDuo™) in a lyophilized, dormant form, which ensures that the bacteria are non-replicating at the time of packaging. The bacteria are reactivated upon contact with water from a separate aqueous phase housed in a dual-chamber delivery system. This enables the bacteria to be applied in a viable form immediately before use.
Importantly, this probiotic serum is already commercially available and has undergone safety and stability evaluations. The formulation strategy and its scientific basis are described in detail in our prior publication:
Podrini C, Schramm L, Marianantoni G, et al. Topical Administration of Lactiplantibacillus plantarum (SkinDuo™) Serum Improves Anti-Acne Properties. Microorganisms. 2023;11(2):417. doi: 10.3390/microorganisms11020417
Reviewer 2 Report
Comments and Suggestions for Authors
SkinDuo TM as a Targeted Probiotic Therapy: Shifts in Skin Microbiota and Clinical Outcomes in Acne Patients
This is a human study that describes the effects of topical application of Lactiplantibacillus plantarum as a probiotic to alter the very diverse bacterial composition that is involved in the phenomenon of acne. Acne is an alteration mainly on the face, with inflammation, erythema, papules, pustules, etc. produced by different endogenous phenomena: age, genetics, hormonal changes and type of diet. These changes in the face are especially annoying and sometimes chronic. And its degree of presentation (from mild to severe) depends on the proliferation of a set of “non-beneficial” bacteria that are often resistant to antibiotics. Therefore, this work describes with graphs, tables and statistics, the variation of the microbiota before and after the application of the probiotic in a group of treated and untreated individuals. The identification of the bacterial flora using molecular techniques is also interesting.
As I have been able to verify after a careful analysis of this work, the results show relative success with this treatment, this study being a preliminary step to continue investigating for a longer time in patients where an improvement has been observed. Above all, to know if this improvement is consolidated over time and if there is a risk of possible toxicity. As the authors propose, acne treatment must be individualized, with this probiotic being another therapeutic resource for the treatment of this multifactorial pathology.
Author Response
We thank the reviewer for their thoughtful comments and for highlighting the relevance of our study. Indeed, this is a human clinical study investigating the effects of the topical application of Lactiplantibacillus plantarum (SkinDuo™ serum) as a probiotic approach to modulate the complex and diverse skin microbiota involved in the pathogenesis of acne.
As correctly noted, acne is a multifactorial skin condition, often chronic, with varying degrees of severity; driven by internal factors such as genetics, hormonal fluctuations, diet, and particularly the proliferation of certain “non-beneficial” bacteria that may also exhibit antibiotic resistance. Our work aims to provide an alternative strategy by restoring microbial balance through the use of beneficial bacteria.
This study evaluates changes in the skin microbiota before and after topical probiotic application, using molecular techniques to profile bacterial populations, and compares treated and untreated groups. The results, presented through graphs, tables, and statistical analysis, indicate a promising trend towards improvement in acne-related symptoms following probiotic use. We agree that this is a preliminary but encouraging step toward longer-term investigations to assess the durability of these effects and the individual variability in response.
Regarding the reviewer’s concern about potential toxicity or irritation, we would like to clarify that a safety assessment was previously conducted. A Skin Irritation Patch Test was independently performed by Complife S.r.l. at the Nutratech headquarters (Cosenza, Italy) on 25 healthy volunteers. The test followed the grading criteria of Berger et al. and found no evidence of erythema, oedema, or irritation in any subjects (Table S1). The mean irritation index was below 0.5, within the acceptable range of 0.25 to 1 for normal skin. These results confirm the safety and tolerability of the SkinDuo™ serum in a healthy population.
We agree with the reviewer’s insight that acne treatment must be individualized. Our findings support the potential of Lactiplantibacillus plantarum as an adjunctive therapeutic option in the personalized management of acne, warranting further long-term studies to evaluate sustained efficacy and safety.
Reference:
Podrini C, Schramm L, Marianantoni G, et al. Topical Administration of Lactiplantibacillus plantarum (SkinDuo™) Serum Improves Anti-Acne Properties. Microorganisms. 2023;11(2):417. doi:10.3390/microorganisms11020417
Reviewer 3 Report
Comments and Suggestions for Authors
In this work, the authors investigate the effects of SkinDuo™, a topical live biotherapeutic containing Lactiplantibacillus plantarum, on the skin microbiota and clinical outcomes in acne patients. They used full-length 16S rRNA sequencing and dermatologist evaluations to identify microbial shifts associated with treatment response. The findings support the potential of microbiome-based therapies for acne. While the study is timely and well-designed, minor revisions are needed to improve the clarity and interpretation of key results. I recommend publication after these issues are addressed.
Comments for the Authors:
- The author should clearly highlight this study's novel aspects and scientific innovation in the introduction section.
- Regarding patient selection, did the authors collect or control for individual variables known to affect skin microbiota and acne? These variables usually include sex, hormonal status, dietary habits, or use of topical products. Please clarify this.
- Was a placebo or vehicle-only control group included to differentiate the effect of SkinDuo™ from general skincare or placebo responses? If not, please discuss this limitation.
- In Figure 3, please review the statistical notation for consistency. It appears that both lowercase “p” and uppercase “P” are used to denote p-values.
- While the clinical grouping into “good” and “no change” responders is mentioned, a quantitative metric would be better to justify this. If it is possible, please specify whether a validated scoring system was used to define treatment response thresholds.
- Please ensure that all references follow a consistent formatting style. Inconsistency is like placing the publication year after the article title. Additionally, the current number of references is insufficient for a primary research article. Please expand the bibliography to appropriately cite relevant prior work.
Author Response
We sincerely thank the reviewer for the thoughtful and constructive feedback. We have carefully addressed each point below and made the appropriate revisions to improve the clarity, scientific rigor, and presentation of our manuscript. Please find our detailed responses below:
-
Novelty and Innovation in the Introduction
Reviewer Comment: The author should clearly highlight this study's novel aspects and scientific innovation in the introduction section.
Response: We thank the reviewer for this important suggestion. We have revised the Introduction to better emphasize the novel aspects and scientific innovation of our study. Specifically, we have clarified these points in lines 51 and 58 of the revised manuscript. -
Patient Selection and Individual Variables
Reviewer Comment: Did the authors collect or control for individual variables known to affect skin microbiota and acne, such as sex, hormonal status, dietary habits, or topical product use?
Response: We appreciate the reviewer’s attention to this crucial methodological detail. This information was indeed collected by the dermatologist during the recruitment process. The majority of participants were female (voluntary recruitment bias), and no subjects were using topical acne products or had polycystic ovary syndrome (PCOS). Dietary habits are summarized in Supplementary Table 2 and are also described in the main text at lines 499 and 505. We acknowledge the gender imbalance as a limitation and thank the reviewer for highlighting it. -
Use of Placebo or Vehicle-Control Group
Reviewer Comment: Was a placebo or vehicle-only control group included? If not, please discuss this limitation.
Response: We thank the reviewer for this important point. A placebo or vehicle-only control was not included in this study, and this has now been clearly stated and discussed as a limitation in lines 455–458. -
Statistical Notation in Figure 3
Reviewer Comment: Please review the statistical notation for consistency.
Response: Thank you for catching this inconsistency. The p-value notation has been corrected for consistency throughout the figure3. -
Clinical Grouping and Response Criteria
Reviewer Comment: Please specify whether a validated scoring system was used to define treatment response thresholds.
Response: We thank the reviewer for this suggestion. The classification into “Good” and “No Change” responders was based on quantitative dermatologist evaluations, as now outlined in Supplementary Table 1 and described in the main text at line 130. A validated scoring framework was followed to ensure objective assessment.Group n Median Pre-Treatment Median Post-Treatment Wilcoxon p-value Good 10 20.5 9 0.00195 No Change 7 21 15 0.109 -
Reference Formatting and Bibliography Size
Reviewer Comment: Please ensure that all references follow a consistent formatting style, and expand the bibliography.
Response: We appreciate this observation. The reference formatting has been revised to ensure consistency, including proper placement of publication years. We have also expanded the bibliography from its initial number to 25 references, incorporating additional recent and relevant studies to strengthen the scientific foundation of our work.
We are grateful for the reviewer’s insightful feedback, which has significantly improved the clarity and quality of our manuscript. We hope the revised version meets the expectations for publication.
Sincerely,
The Authors